# The Formyl Peptid Receptor Ligand Ac2-26 Improves the Integrity of the Blood−Brain Barrier in the Course of Pneumococcal Meningitis

**DOI:** 10.3390/cells13242104

**Published:** 2024-12-19

**Authors:** Johannes Deutloff, Irina Pöhner, Johann Rößler, Markus Kipp, Simone C. Tauber, Lars-Ove Brandenburg

**Affiliations:** 1Institute of Anatomy, Rostock University Medical Center, Rostock, Gertrudenstraße 9, 18057 Rostock, Germanymarkus.kipp@med.uni-rostock.de (M.K.); 2Department of Neurology, RWTH University Hospital Aachen, Pauwelsstraße 30, 52074 Aachen, Germany

**Keywords:** formyl peptide receptor, bacterial meningitis, blood-brain barrier, innate immunity, *Streptococcus pneumoniae*, annexin A1

## Abstract

Background: The brain is protected from invading pathogens by the blood−brain barrier (BBB) and the innate immune system. Pattern recognition receptors play a crucial role in detecting bacteria and initiating the innate immune response. Among these are G-protein-coupled formyl peptide receptors (FPR), which are expressed by immune cells in the central nervous system. In this study, we investigated the influence of the FPR ligand Ac2-26 on the integrity of the BBB during pneumococcal meningitis. Methods: Wild-type (WT) and Fpr1- and Fpr2-deficient mice were intrathecally infected with *Streptococcus pneumoniae*. Subsequently, different groups of mice were treated with intraperitoneal injections of Ac2-26. The integrity of the BBB was analyzed using various markers through immunohistochemistry and immunofluorescence. Results: The results showed reduced BBB integrity during the course of bacterial meningitis. Treatment with Ac2-26 in WT mice significantly prolonged the maintenance of BBB integrity. However, this effect was not observed in Fpr2-deficient mice. Conclusions: This study extends previous findings on the anti-inflammatory properties of Ac2-26 by demonstrating that Ac2-26 positively affects BBB integrity via FPR2 during pneumococcal meningitis. These findings suggest that further investigation of Ac2-26 and other FPR modulators as potential therapies for *Streptococcus pneumoniae*-induced meningitis is warranted.

## 1. Introduction

Bacterial meningitis induces a strong acute inflammation reaction in the central nervous system (CNS). Despite prompt antibiotic therapy, the disease has a high mortality rate, and many patients experience long-term sequelae after recovery. Up to 37% of affected individuals die as a result of the disease [1]. After recovery, up to half of the patients experience long-term sequelae [2,3]. Mortality primarily depends on the pathogen, age of the patient, and timing of antibiotic therapy initiation. The most common causes of bacterial meningitis are the Gram-negative meningococci *Neisseria meningitidis* and the Gram-positive pneumococci *Streptococcus pneumonia*, which account for about 80% of bacterial meningitis cases in older children and adults [1]. The bacteria can enter the brain per continuitatem, but they have also evolved mechanisms to penetrate the brain by crossing the blood−brain barrier (BBB). The BBB is formed by endothelial cells, the basal membrane and the pericytes. Additionally, astrocytes form a perivascular sheath around the blood vessels with their end-feet. The endothelial cells in the brain differ from peripheral endothelial cells in several ways. They have specialized tight junctions (TJ) that largely prevent paracellular transport. TJ consists of four different protein classes: occludins, claudins, tricellulins, and Junctional Adhesion Molecules (JAM). Another connection between endothelial cells is the adherens junction, which is primarily composed of vascular endothelial (VE)-cadherin. This endothelial cell binding is regulated and stabilized by platelet/endothelial cell adhesion molecule-1 (PECAM-1) [4,5].

Once the bacteria have reached the subarachnoid space, they can multiply rapidly and spread further into the brain parenchyma. This triggers a massive inflammatory response, including the migration of peripheral immune cells such as neutrophil granulocytes, as well as the activation of innate immune cells in the brain, including microglia cells and astrocytes. These immune cells can recognize a wide range of broadly defined molecular motifs in pathogens. These molecular determinants, termed pathogen-associated molecular patterns (PAMPs) or danger-associated molecular patterns (DAMPs), are conserved molecular patterns recognized by so-called pattern recognition receptors (PRRs). These receptors, which include Toll-like receptors and G-protein-coupled formyl peptide receptors (FPRs), are located in both the plasma membrane and intracellular environments. While three representatives of the FPR gene family have been identified in humans to date, at least six representatives, including the two main representatives Fpr1 and Fpr2, have been identified in the mouse genome [6,7]. In the CNS, the two representatives could be characterized by the glial cells [8,9].

Our previous research on the role of FPRs in bacterial meningitis showed that mice deficient in mFPR1 or mFPR2 experience a more severe course of pneumococcal meningitis in terms of survival, bacterial load, and inflammatory response compared to wild-type mice [10]. Interestingly, FPRs are characterized by a multitude of ligands, which include both endogenous and exogenous ligands [6,11]. These ligands are associated with various species of different diseases and include pro- and anti-inflammatory molecules [12,13,14,15]. One well-known anti-inflammatory ligand of FPRs is the endogenous protein annexin A1 or its N-terminal fragment Ac2-26 [16]. Ac2-26 has been shown to inhibit leukocyte adhesion and transmigration, thereby limiting the intensity and duration of the inflammatory response, while promoting epithelial cell proliferation and migration [17,18]. Beyond, Annexin A1 and its fragment Ac2-26 have demonstrated positive effects in various preclinical models of pathological conditions, such as multiple sclerosis, pneumococcal pneumonia, arthritis, uveitis, and wound healing [19,20,21]. Moreover, a recent study showed that treatment with Ac2-26 reduced the inflammatory response following pneumococcal meningitis in an FPR2-dependent manner [15].

In this study, we investigated the effects of the FPR agonist Ac2-26 on the integrity of the BBB during pneumococcal meningitis. We used a mouse model of pneumococcal meningitis and demonstrated that treatment with Ac2-26 significantly prolonged the integrity of the BBB following pneumococcal meningitis. These results extend and complement previous findings on the positive effects of this anti-inflammatory FPR ligand.

## 2. Materials and Methods

### 2.1. Reagents

The peptide Ac2-26 (Ac-AMVSEFLKQAWFIENEEQEYVQTVK), obtained from Bankpeptide Limited (Hefei, China), was prepared in phosphate-buffered saline (PBS). Mice received intraperitoneal injections of Ac2-26 at a dose of 1 mg per kilogram of body weight at 2, 8, and 24 h after infection.

### 2.2. Animals

Fpr1-KO, Fpr2-KO, and wild-type (WT) mice were housed at the Central Animal Care Facility of the RWTH Aachen University. Fpr1-KO mice were kindly provided by Dr. Philip Murphy from the National Institute for Allergy and Infectious Diseases, NIH, Bethesda, MD [7]. Fpr2-KO mice were developed as outlined in previous research [22]. Both knockout strains were bred on a C57BL/6 genetic background. Wild-type (WT) mice, used as controls, were backcrossed onto a C57BL/6 J background for at least five generations. To minimize the influence of hormonal variability, only male mice were included in the experiments.

### 2.3. Induction of Experimental Pneumococcal Meningitis

All animal experiments were approved by the Animal Care Committee of the University Hospital Aachen and the District Government of Recklinghausen, North Rhine-Westphalia, Germany (approval number: 84-02.04.2015.A157). The procedures were carried out in compliance with the ARRIVE guidelines for the ethical use of laboratory animals [23]. The pneumococcal meningitis model employed in this study has been described in earlier publications [24,25]. In brief, 8-week-old WT, Fpr1-KO, and Fpr2-KO mice were anesthetized with ketamine (100 mg/kg body weight) and xylazine (20 mg/kg body weight). Mice were infected with the *Streptococcus pneumoniae* D39 strain (type 2) at a concentration of 10^4^ colony-forming units/mL via injection into the subarachnoid space through the skull. Control animals received sterile saline instead. Signs of infection developed in the infected mice within the first 24 h. For histological examination, mice were euthanized at various post-infection time points (n ≥ 4 per group). Unless otherwise specified, the experiments were repeated three times independently. At the end of each experiment, the mice were sacrificed and perfused with 4% formalin for immunohistochemical and immunofluorescence analyses.

### 2.4. Immunohistochemistry or Immunofluorescence

For immunohistochemistry, tissue sections were first rehydrated, and antigen retrieval was carried out using a Tris/EDTA buffer with a pH of 9.0. The sections were then rinsed with PBS and incubated overnight at 4 °C with primary antibodies diluted in a blocking solution containing serum from the species used to produce the secondary antibody. The primary antibody used was anti-fibrinogen (1:2000; NBP1-90956; Novus Biologicals, Wiesbaden, Germany). The following day, sections were treated with biotinylated secondary antibodies (1:50; BA-1000; Biozol, Eching, Germany) for 1 h. After additional PBS washes, the sections were incubated with peroxidase-labeled avidin-biotin complex (ABC kit; Vector Laboratories, Peterborough, UK). Antigen-antibody complexes were visualized using 3,3′-diaminobenzidine (DAKO, Hamburg, Germany) as the peroxidase substrate. Finally, the slides were counterstained with hematoxylin and mounted using DePeX (Serva, Heidelberg, Germany). For immunofluorescence staining, slides were incubated with anti-CD31 (1:50; ab28364, Abcam, Cambridge, UK) or anti-Claudin-5 (1:20; 35-2500, Thermo Fisher Scientific, Dreieich, Germany) antibodies, followed by incubation with anti-rabbit or anti-mouse IgG Alexa Fluor 594 secondary antibodies (1:250; A21207 or A21203, Thermo Fisher Scientific, Dreieich, Germany). To visualize cell nuclei, sections were incubated with bisbenzimide (1:10,000 in PBS) and mounted on ImmuMount (Thermo Fisher Scientific).

### 2.5. Quantification of the Optical Density

Brain sections immunohistochemically stained for fibrinogen were analyzed by determining the optical density. The sections were digitized using a microscope scanner (Grundium, Ocus 20, Tampere, Finland). Optical density was determined using the “QuPath” software (version 0.3.2 for Mac OS X, University of Edinburgh, Edinburgh, UK). To this end, images were imported into the software environment, and the entire brain area was manually outlined using the “polygon selection” tool. A “threshold” was used for the analysis. Settings were defined on the first image to best match the immunohistochemical staining pattern and were then retained throughout the analysis. The following settings were used: “Resolution”: High (1.00 µm/px), “Channel”: DAB, “Prefilter”: Gaussian, “Smoothing sigma”: 1, “Threshold”: 0.5. Values above the defined threshold of 0.5 were considered positive, while those below were negative. The software calculated the positive area as a percentage of the entire outlined area.

### 2.6. Quantification of Fluorescence Intensity

Brain sections stained by immunofluorescence were analyzed by determining the fluorescence intensity. Therefore, the sections were digitized using a microscope (Leica DM6 B, Wetzlar, Germany) equipped with a camera system (Leica DMC 6200 Wetzlar, Germany). Intensity was measured using “ImageJ” software (version 1.53k for Mac, Wayne Rasband, and National Institutes of Health, Bethesda, MD, USA). The images were imported into the program, and vessels were manually outlined using the “polygon selection” tool. The mean gray value was then calculated.

### 2.7. Statistical Analyses

All data are presented as arithmetic means ± SEM. Differences between groups were statistically tested using GraphPad Prism 10 software (GraphPad Software Inc., San Diego, CA, USA). D’Agostino and Pearson tests were used to assess the Gaussian distribution of the data. Statistical comparisons between groups were performed using two-way ANOVA followed by Bonferroni’s post-hoc multiple comparison test. The details of the statistical methods used for each analysis are included in the main text and figure legends. A *p*-value of ≤0.05 was considered statistically significant. The levels of significance are indicated as follows: * *p* ≤ 0.05, ** *p* ≤ 0.01, *** *p* ≤ 0.001, and ns = not significant.

## 3. Results

### 3.1. Acute Bacterial Meningitis Induces Breakdown of the Blood−Brain Barrier

In the first step, we investigated the time course of BBB permeability by assessing the presence of typical BBB markers. First, the effect of bacterial meningitis was analyzed based on fibrinogen permeability. Fibrinogen is a glycoprotein with a molecular weight of approximately 340 kDa that is produced in the liver and plays a central role in blood coagulation. Under physiological conditions, it is absent in the brain and only penetrates brain tissue when the BBB integrity is compromised [26]. We, therefore, measured the positive area for fibrinogen in the brains of *Streptococcus pneumoniae*-infected mice at different time points. As shown in Figure 1, infection led to a time-dependent increase in fibrinogen-positive areas, with the maximum observed in the mortality group, while the non-infected group showed almost no fibrinogen-positive staining (Figure 1A). This was confirmed by image analysis, where the mortality group showed the highest average value of 9% positive area, while the non-infected mice had the lowest, averaging 0.018% (Figure 1B; ** *p* < 0.01; one-way ANOVA followed by Bonferroni test).

Next, we analyzed PECAM-1 (CD-31), a vascular cell adhesion and signaling molecule that acts as an adhesive stress-response protein in the BBB and influences its integrity and recovery after inflammatory changes [27]. Pneumococcal adhesins bind to PECAM-1, potentially mediating *Streptococcus pneumoniae* penetration through the BBB [28]. As shown in Figure 2A, the non-infected brain slices displayed very weak immunofluorescence staining for PECAM-1, while the strongest increase was detected 48 h after infection. Time-course evaluation revealed a maximum immunofluorescence signal at 48 h post-infection (mean fluorescence intensity of 19.39, based on the mean gray values), followed by a decrease in the mortality group. The increase in fluorescence intensity at 48 h was significant compared to that in all other groups (Figure 2B; * *p* < 0.05, one-way ANOVA followed by Bonferroni test). Interestingly, an increase in the number of nuclei around the vessels was also observed, suggesting an influx of infiltrating cells.

Lastly, claudin-5, an essential protein for tight junction formation in the BBB, was examined [29]. As shown in Figure 3A, the highest immunofluorescence signal was detected in the capillary walls of the control group, with the signal decreasing over the course of infection. This was confirmed by quantifying the fluorescence intensity. The reduction in claudin-5 fluorescence intensity at 48 h post-infection and in the mortality group was significant (Figure 3B; ** *p* < 0.01, *** *p* < 0.001; one-way ANOVA followed by Bonferroni test). As in Appendix A, we have summarized the results of Figure 1, Figure 2 and Figure 3, including any significant differences that were highlighted.

### 3.2. Lack of FPR1 Changes Permeability of BBB in Uninfected Controls

Our previous studies have demonstrated that the absence of FPRs alters the intensity and efficacy of the immune response during pneumococcal meningitis. Moreover, the anti-inflammatory FPR ligand Ac2-26 reduces meningeal granulocyte infiltration and bacterial load during pneumococcal meningitis [10,15]. Therefore, we investigated the influence of FPRs and the FPR ligand Ac2-26 on BBB permeability. First, we quantified the fibrinogen-positive area in control mice (without infection) with and without Ac2-26 treatment. As shown in Figure 4A, differences between untreated and treated WT, FPR1-KO, and FPR2-KO controls were observed but were not statistically significant (*p* > 0.05). Next, PECAM-1 immunofluorescence analysis was performed (Figure 4B). Interestingly, FPR1-KO control mice exhibited a significant increase in PECAM-1 immunofluorescence compared to WT controls (* *p* < 0.05). In Ac2-26-treated FPR1-KO control mice, the PECAM-1 signal increased even further (** *p* < 0.01; two-way ANOVA followed by the Bonferroni test).

Lastly, we analyzed claudin-5 fluorescence intensity in treated WT, FPR1-KO, and FPR2-KO control mice with or without Ac2-26 treatment. As shown in Figure 4C, FPR1-KO control mice displayed a significant reduction in claudin-5 fluorescence intensity compared to WT or FPR2-KO controls (* *p* < 0.05; two-way ANOVA followed by Bonferroni test). The reduction in fluorescence intensity in all three genotypes by Ac2-26 was not significant.

### 3.3. Ac2-26 Ameliorates BBB Permeability Loss During Pneumococcal Infection

Our previous findings showed that FPR2 tends to ameliorate the inflammatory response, while FPR1 activation exacerbates inflammation [15]. We therefore compared WT, FPR1-, and FPR2-KO infected, either infected or uninfected, with or without Ac2-26 treatment 30 h after pneumococcal infection. At this point, the infected mice show a moderate inflammatory reaction, and there is no danger that some of the mice are already very ill or have already died [15]. First, we again measured the fibrinogen-positive area per brain. As in Figure 4A, no differences between the controls with or without Ac2-26 were identified; we included only one control for fibrinogen in Figure 5A. As shown in Figure 5A, in WT mice, fibrinogen deposition in the CNS parenchyma increased in infected mice, and this increase was reduced by Ac2-26 treatment. In FPR1-KO mice, the same fibrinogen staining pattern was observed, with increased staining intensities in infected mice, which were reduced by Ac2-26 treatment. Of note, Ac2-26 treatment did not ameliorate fibrinogen deposition in FPR2-KO mice, suggesting that this Ac2-26 effect is mediated by FPR2 and not FPR1.

Next, PECAM-1 immunofluorescence was compared across genotypes and after Ac2-26 treatment during infection. The changes in PECAM-1 immunofluorescence intensity in infected WT mice with or without Ac2-26 were not significant. (Figure 5B). Interestingly, FPR1-KO mice displayed strong PECAM-1 immunofluorescence in the controls. Infection led to a decrease in PECAM-1 signal in FPR1-KO mice, but Ac2-26 treatment significantly increased PECAM-1 immunofluorescence in infected FPR1-KO mice (*p* < 0.01).

Finally, claudin-5 fluorescence intensity was compared. As shown in Figure 5C, infection led to a significant reduction in claudin-5 fluorescence intensity in WT and FPR2-KO mice (**** *p* < 0.0001; two-way ANOVA followed by the Bonferroni test). Ac2-26 treatment in infected WT and FPR1-KO mice increased claudin-5 fluorescence intensity, although the changes were not significant. In FPR2-KO mice, no difference was observed between infected mice in the presence or absence of Ac2-26 treatment. As in Appendix A, we have summarized the results of Figure 4 and Figure 5, including any significant differences that were highlighted.

## 4. Discussion

In this study, we investigated the influence of FPRs and the FPR ligand Ac2-26 on the integrity of the BBB during pneumococcal meningitis. Mice were infected with *Streptococcus pneumoniae* and compared at different time points, including the mortality group. The course of infection is accompanied by massive bacterial growth in the cerebrospinal fluid, followed by significant infiltration of the CNS by neutrophil granulocytes and a strong inflammatory reaction of the CNS tissue due to activation of glial cells [30].

This process is associated with BBB breakdown [31]. The formation of the BBB by brain microvascular endothelial cells (BMECs) involves the basement membrane, astrocytes, and pericytes, with numerous tight junctions (TJs) and adherens junctions between neighboring cells responsible for the BBB’s limited permeability [31]. Our results show that the BBB collapses over time, leading to increased permeability. The concentration of the inflammatory marker fibrinogen increases steadily in the brain tissue, indicating vascular dysfunction (Figure 1). In contrast, the TJ protein claudin-5 decreases progressively as the infection advances (Figure 3). Claudin-5, one of the most abundant TJ proteins, is crucial for size selectivity of the BBB [32]. Disruption or reduction of claudin-5 may be involved in the pathophysiology of various diseases, including schizophrenia, depression, and Alzheimer’s disease [33,34,35]. In other infection models, such as infection with Group B Streptococcus, CNS tuberculosis, or viral infections of the CNS, the importance of claudin-5 for the course of infection and bacterial passage through the BBB was also demonstrated [36,37,38]. Regarding PECAM-1, our results show a peak in expression at 48 h, while the mortality group exhibited a decrease (Figure 2). PECAM-1 acts as a stress-response protein, influencing BBB integrity and the restoration of the barrier after an inflammatory response [27]. PECAM-1 also plays a key role in leukocyte migration from vessels into surrounding tissue, suggesting it has anti-inflammatory properties [39]. Regarding PECAM-1, our results show a peak in expression at 48 h, while the mortality group exhibited a decrease (Figure 2). PECAM-1 acts as a stress-response protein, influencing BBB integrity and the restoration of the barrier after an inflammatory response [28,40]. An increase in PECAM-1 during infection can increase bacterial load in the brain, supporting infection progression, as shown in previous studies [15]. The opening of the BBB and increase in PECAM-1 allow more bacteria and peripheral immune cells to enter the CNS, causing further damage.

The breakdown of the BBB can result from various processes; however, pneumolysin, released through the autolysis of *Streptococcus pneumoniae*, plays a central role [41]. Pneumolysin interacts with the cell membranes of brain microvascular endothelial cells, creating pores that disrupt membrane integrity, ultimately leading to cell lysis [42]. Pneumolysin also induces astrocyte shrinkage, facilitating the spread of bacteria and toxins within the CNS [43]. In addition to its impact on astrocytes, pneumolysin inhibits microglial cell motility [44]. In patients who died from streptococcal meningitis, higher levels of pneumolysin were found in the cerebrospinal fluid compared to survivors, likely reflecting the bacterial load [45]. Studies have also shown that pneumolysin causes neuronal damage, possibly linked to increased intracellular calcium levels, which activates apoptosis-inducing factors [46,47]. Neuronal damage and apoptosis increase the risk of dementia [48,49]. In many neuroinflammatory or neurodegenerative diseases, the loss of BBB integrity and the resulting neuroinflammation leads to severe CNS damage, contributing to disease progression [50].

The immune system must be activated to recognize and defend against pathogens. The innate immune system and its pattern recognition receptors play a central role in this process [51]. In addition to Toll-like receptors, pattern recognition receptors also include formyl peptide receptors (FPRs). FPRs respond to a broad range of ligands that can originate from both pathogens and host cells. These include both pro- and anti-inflammatory ligands, suggesting that FPRs may play a role in resolving inflammatory responses [12,17]. Our previous research on FPRs in the context of pneumococcal meningitis demonstrated that FPR deficiency leads to a more severe disease course, including increased bacterial burden, heightened neutrophil infiltration, and higher mortality [10]. Treatment with the N-terminal bioactive annexin A1 fragment Ac2-26, an FPR ligand, reduces neutrophil recruitment and bacterial load, accompanied by decreased glial cell activation and inflammation [15]. Our current results show that even non-infected FPR1-deficient mice exhibit significantly higher PECAM-1 expression and lower claudin-5 expression compared to controls (Figure 4). This aligns with our earlier findings that FPR1-KO mice have higher bacterial titers than WT and FPR2-KO mice [10,15]. BBB permeability decreases during infection, but treatment with Ac2-26 ameliorates this breakdown. This supports our previous finding that Ac2-26 treatment leads to reduced bacterial titers and neutrophil infiltration [15]. Annexin A1 has been shown to modulate neutrophil adhesion to the vessel wall, inhibiting adhesion to endothelial monolayers and transmigration [52,53,54]. In addition, annexin A1 reduces myeloperoxidase (MPO) activity during *Streptococcus suis* serotype 2 meningitis about the FPR2 [55]. For the MPO, it was previously shown that inhibition improves the integrity of the BBB [56]. In diseases such as conjunctivitis, uveitis, peritonitis, and pneumonia, annexin A1 or Ac2-26 reduces neutrophil recruitment [18,57,58,59]. Previous studies have shown that annexin A1 and its fragment Ac2-26, have a high affinity for FPR2 [15,16,53]. Our earlier results demonstrated that the anti-inflammatory properties of Ac2-26 in a model of pneumococcal meningitis are mediated via FPR2 but not FPR1 [15]. The current results support the anti-inflammatory role of Ac2-26, showing that it tends to stabilize the BBB. Ac2-26 tends to reduce fibrinogen levels in brain tissue and increase claudin-5 expression. These effects were observed in infected WT and FPR1-KO mice, but not in infected FPR2-KO mice.

## 5. Conclusions

In conclusion, our findings enhance the understanding of the role of FPRs and their ligands in pneumococcal meningitis. The absence of one of the FPRs has a detrimental effect on the inflammatory response, underscoring the importance of a balanced immune response. Additionally, Ac2-26 improves CNS protection by reducing inflammation and enhancing BBB integrity, thus offering a potential therapeutic approach. Further exploration of FPR modulation during inflammation could be a promising area for future research.

## Figures and Tables

**Figure 1 cells-13-02104-f001:**
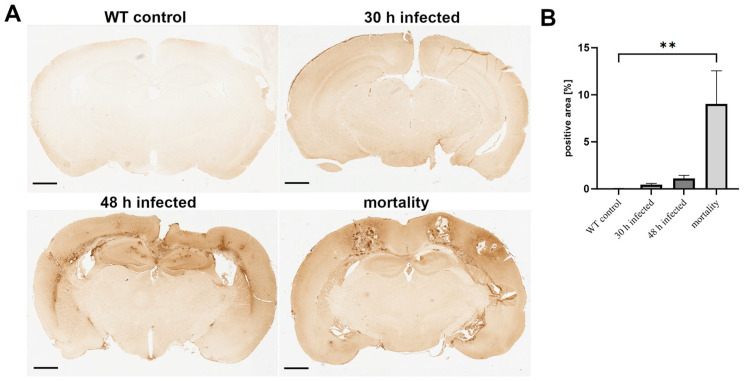
Increased fibrinogen infiltration during the course of pneumococcal meningitis. Wild-type (WT) mice were infected by administering 10^4^ colony-forming units (CFU) of *Streptococcus pneumoniae* D39 type 2 strain into the subarachnoid space. At 30 and 48 h post-infection, and in the mortality group, the fibrinogen-positive area was visualized using an antibody. (**A**) The images show representative results. Scale bar: 800 µm. (**B**) Quantification of the fibrinogen-positive area [%] (WT control, n = 5; 30 h post-infection, n = 5; 48 h post-infection, n = 4; mortality group, n = 4). One-way ANOVA followed by Bonferroni test (**—*p* < 0.01).

**Figure 2 cells-13-02104-f002:**
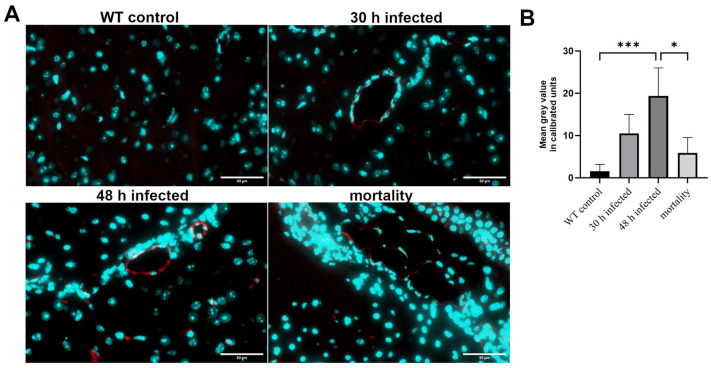
Changes in PECAM-1 immunofluorescence during the course of pneumococcal meningitis. Wild-type (WT) mice were infected by administering 10^4^ colony-forming units (CFU) of *Streptococcus pneumoniae* D39 type 2 strain into the subarachnoid space. At 30 and 48 h post-infection, as well as in the mortality group, PECAM-1 immunofluorescence was visualized using an anti-PECAM-1 antibody and detected with an anti-rabbit anti-Cy3 secondary antibody (in red; cell nuclei in blue). (**A**) The images show representative results. Scale bar: 50 µm. (**B**) Quantification of the PECAM-1-positive area [%] (WT control, n = 5; 30 h post-infection, n = 5; 48 h post-infection, n = 4; mortality group, n = 4). One-way ANOVA followed by Bonferroni test (*—*p* < 0.05; ***—*p* < 0.001).

**Figure 3 cells-13-02104-f003:**
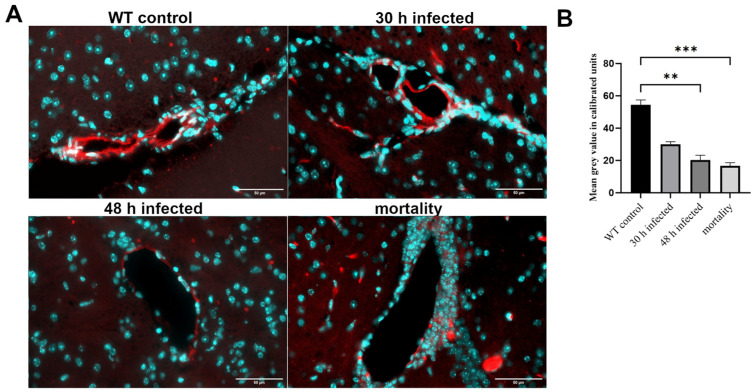
Decrease in Claudin-5 immunofluorescence during the course of pneumococcal meningitis. Wild-type (WT) mice were infected by administering 10^4^ colony-forming units (CFU) of the *Streptococcus pneumoniae* D39 type 2 strain and were introduced into the subarachnoid space. At 30 and 48 h post-infection, as well as in the mortality group, Claudin-5 immunofluorescence was visualized using an anti-Claudin-5 antibody and detected with an anti-mouse anti-Cy3 secondary antibody (in red; cell nuclei in blue). (**A**) The images show representative results. Scale bar: 50 µm. (**B**) Quantification of Claudin-5-positive area [%] (WT control, n = 5; 30 h post-infection, n = 5; 48 h post-infection, n = 4; mortality group, n = 4). One-way ANOVA followed by Bonferroni test (**—*p* < 0.01; ***—*p* < 0.001).

**Figure 4 cells-13-02104-f004:**
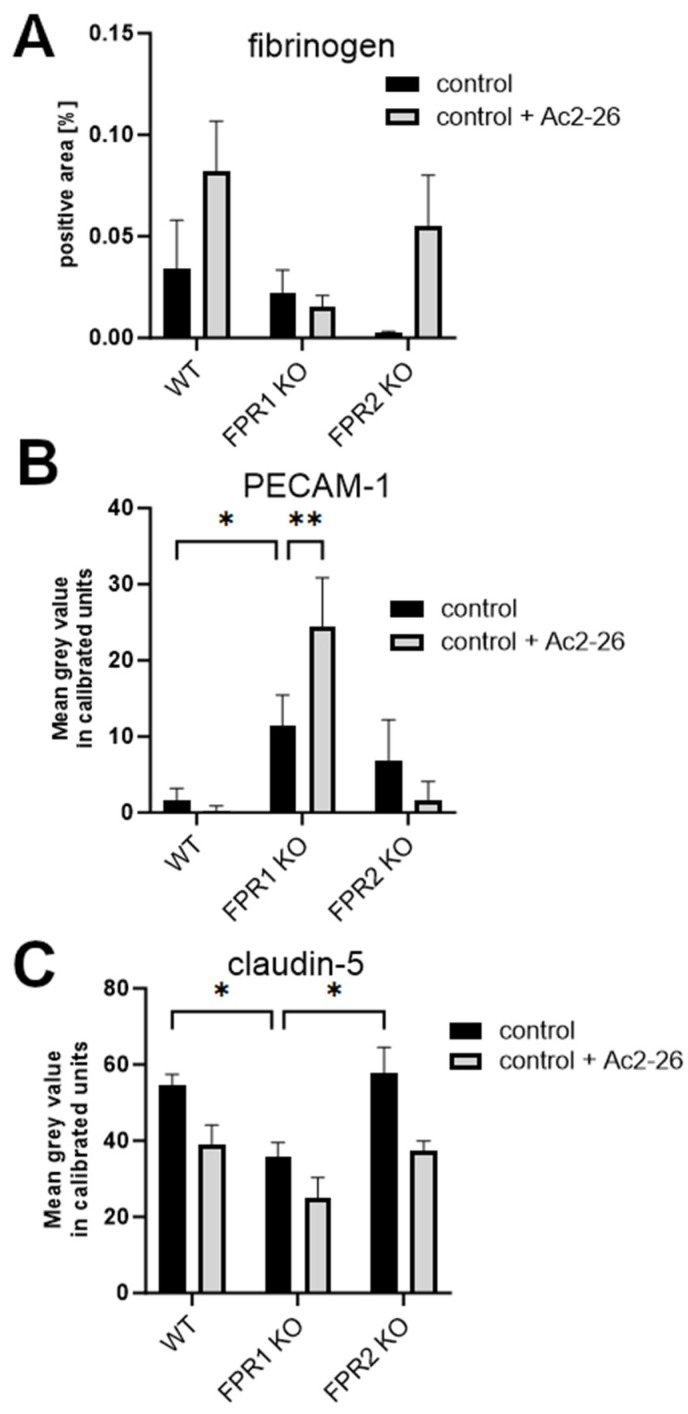
Lack of FPR1 alters BBB permeability in uninfected controls. (**A**) Quantification of fibrinogen-positive area, (**B**) PECAM-1, or (**C**) Claudin-5 immunofluorescence in wild-type (WT) and formyl peptide receptor 1 (FPR1) or FPR2 knockout (KO) mice (WT control, n = 7; WT control + Ac2-26, n = 5; FPR1 KO control, n = 4; FPR1 KO control + Ac2-26, n = 4; FPR2 KO control, n = 4; FPR2 KO control + Ac2-26, n = 4). One-way ANOVA followed by Bonferroni test (*—*p* < 0.05; **—*p* < 0.01).

**Figure 5 cells-13-02104-f005:**
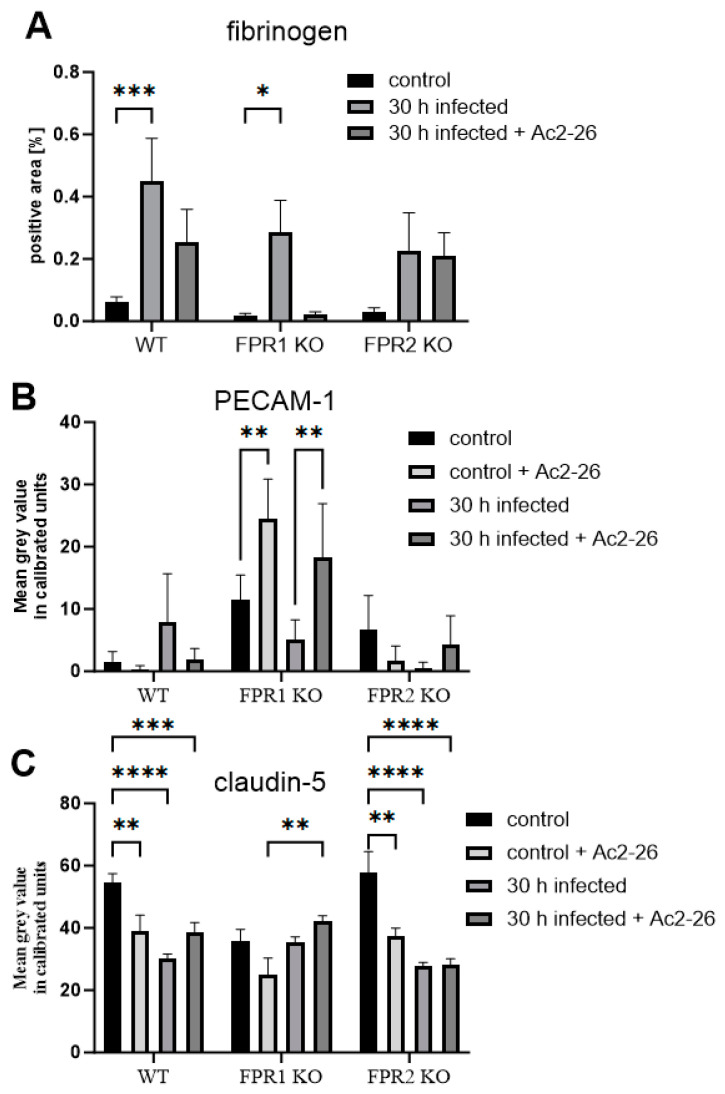
Ac2-26 mitigates BBB permeability loss during pneumococcal infection. (**A**) Quantification of fibrinogen-positive area, (**B**) PECAM-1, or (**C**) Claudin-5 immunofluorescence in wild-type (WT) and formyl peptide receptor 1 (FPR1) or FPR2 knockout (KO) mice, 30 h post-infection with 10^4^ colony-forming units (CFU) of the *Streptococcus pneumoniae* D39 type 2 strain were introduced into the subarachnoid space (WT control, n = 7; WT control + Ac2-26, n = 5; WT 30 h infected, n = 9; WT 30 h infected + Ac2-26, n = 7; FPR1 KO control, n = 4; FPR1 KO control + Ac2-26, n = 4; FPR1 KO 30 h infected, n = 5; FPR1 KO 30 h infected + Ac2-26, n = 7; FPR2 KO control, n = 4; FPR2 KO control + Ac2-26, n = 4; FPR2 KO 30 h infected, n = 5; FPR2 KO 30 h infected + Ac2-26, n = 6). One-way ANOVA followed by the Bonferroni test (*—*p* < 0.05; **—*p* < 0.01; ***—*p* < 0.001; ****—*p* < 0.0001).

## Data Availability

The authors confirm that the data supporting the findings of this study are available within the article.

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
