# Peer review of "The Formyl Peptid Receptor Ligand Ac2-26 Improves the Integrity of the Blood−Brain Barrier in the Course of Pneumococcal Meningitis"

_cells, 2024, doi:10.3390/cells13242104_

Round 1

Reviewer 1 Report

Comments and Suggestions for Authors

This study investigated the possible roles of formyl peptide receptors (FPRs) - which recognize bacterial PAMPs and DAMPs - in blood-brain-barrier breakdown in a mouse model of pneumococcal meningitis, using WT, Fpr-1 and Fpr-2 knock out mice, and the Fpr agonist Ac2-26 in various combinations.  Specifically, changes in fibrinogen deposition in brain, and claudin-5 and PECAM-1 immunofluorescence were assessed at a series of time points following intrathecal injection of S pneumococcus, with or without IP injection of Ac2-26. A variety of changes were seen over time but the description of them is complex and extremely challenging to follow, particularly since broad statements are made but only some of the actual data are presented, much of that in graphic or visual form.

I have 3 overarching recommendations to the. authors

1.        Please do not describe changes in values in specific experiments and at the end of the sentence state these were not statistically significant.  It makes it even more difficult to follow the logic

2.         Please include a Table summarizing all the data – infected, control, WT, Fpr-1-KO, Fpr-2-KO, with and without Ac2-26 and the various time points, indicating which are statistically significant

3.        In the discussion, rather than discussing possible roles of these pathways in schizophrenia, depression and Alzheimer’s, focus on what this actually does and does not show in CNS infection

Author Response

Reviewer 1

  1. Please do not describe changes in values in specific experiments and at the end of the sentence state these were not statistically significant. It makes it even more difficult to follow the logic

We appreciate the reviewers’ valuable feedback and have revised the manuscript accordingly to address the comments and provide greater clarity.

  1. Please include a Table summarizing all the data – infected, control, WT, Fpr-1-KO, Fpr-2-KO, with and without Ac2-26 and the various time points, indicating which are statistically significant

As the supplemental table 1 and 2. we have summarised the results of the different figures including any significant differences that were highlighted.

  1. In the discussion, rather than discussing possible roles of these pathways in schizophrenia, depression and Alzheimer’s, focus on what this actually does and does not show in CNS infection

In line with the reviewer’s suggestion, we have adjusted the discussion to focus on what our findings demonstrate specifically about CNS infections. We have reduced references to the potential roles of these markers or the BBB in schizophrenia, depression, and Alzheimer’s disease. Instead, the discussion now emphasizes the relevance of our findings to CNS infections, clarifying their implications and limitations in this context.

Reviewer 2 Report

Comments and Suggestions for Authors

This article is relatively reasonable and reliable. The aim of this study was investigating the influence of an FPR ligand Ac2-26 on the integrity of the BBB during pneumococcal meningitis by various markers through immunohistochemistry and immunofluorescence. The article ’s logic is rigorous, and the data of the experiments are convincing. The authors unique findings showed treatment with Ac2-26 in WT mice significantly prolonged the maintenance of BBB integrity, however, this effect was not observed in FPR2-deficient mice. It is a topic of interest to the researchers in the related areas, but the article still has some problems.

1. Please provide the chemical structure of Ac2-26.

2. The mechanisms of Ac2-26 prolonged the integrity of the BBB following pneumococcal meningitis need to be proved in this research instead of only proving by phenotype.

3. Activity experiments are best performed by selecting a positive control drug for drug effect comparisons.

Author Response

Reviewer 2

  1. Please provide the chemical structure of Ac2-26.

We appreciate the reviewer’s request for further clarification. The amino acid sequence of Ac2-26 has been listed in the Reagents section of the Materials and Methods. Additionally, to address this comment, we have now included a more precise representation of the chemical structure of Ac2-26 based on the figure. This information has been added to the manuscript for enhanced clarity.Ac2-26:

M. Wt

3089.46

Formula

C141H210N32O44S

Sequence

AMVSEFLKQAWFIENEEQEYVQTVK (Modifications: Ala-1 = N-terminal Ac)

N-acetyl-L-alanyl-L-methionyl-L-valyl-L-seryl-L-α-glutamyl-L-phenylalanyl-L-leucyl-L-lysyl-L-glutaminyl-L-alanyl-L-tryptophyl-L-phenylalanyl-L-isoleucyl-L-α-glutamyl-L-asp

  1. The mechanismsof Ac2-26 prolonged the integrity of the BBB following pneumococcal meningitis need to be proved in this research instead of only proving by phenotype.

We agree that elucidating the mechanisms by which Ac2-26 influences BBB integrity is an important and valid point. However, there is currently limited preliminary research on this topic in the literature, particularly in the context of bacterial meningitis.

For Ac2-26, there are no established findings regarding its direct effects on BBB integrity during bacterial meningitis. However, for annexin A1, it has been demonstrated that its application reduces the adhesion of neutrophil granulocytes to the endothelial cell line bEnd.3. Additionally, during Streptococcus suis serotype 2 meningitis, annexin A1 treatment was shown to reduce myeloperoxidase (MPO) activity (DOI: 10.1128/IAI.00680-20). Given that MPO reduction or inhibition has been associated with improved BBB integrity (DOI: 10.1111/jnc.13426), this pathway may offer a potential explanation for the effects observed with Ac2-26 in our study.

While the exact mechanisms require further exploration, our current manuscript provides a significant initial indication of Ac2-26's potential to maintain BBB integrity in the context of pneumococcal meningitis. We are actively pursuing further investigations, including measurements of transendothelial electrical resistance (TEER) in vitro and FITC-dextran permeability assays in vivo, to better elucidate the underlying mechanisms. However, these results are not yet available and should be considered preliminary at best.

To address this point, we have now included a discussion of the known effects of annexin A1 in the context of bacterial meningitis in the revised manuscript. We hope this addition provides further context and supports the importance of this area of research.

  1. Activity experiments are best performed by selecting a positive control drug for drug effect comparisons.

Thank you for the comment. That is an important point. However, it is difficult to specifically name a positive control here. Many FPR ligands are known to be pro-inflammatory (e.g. fMLF as a bacterial cell wall component) or anti-inflammatory ligands such as annexin A1 and its fragment Ac2-26. No findings on other FPR ligands apart from annexin A1/Ac2-26 can be found in the literature. We have tested fMLF, Ac2-26 and an antagonist, Boc2, in the context of Alzheimer's disease. However, the influence on the BBB was not investigated here (https://doi.org/10.1186/s12974-020-01816-2). Interestingly, the FPR antagonist Boc2 showed the best effect on the amyloid beta-induced inflammatory reaction, while Ac2-26 showed no improvement. We have just started a project where we are looking at the effect of ligands on the BBB in Alzheimer's disease, but we cannot show any results yet. Furthermore, in the context of bacterial meningitis, it would be interesting to test and compare different annexin A1 derivatives or other anti-inflammatory FPR ligands. But this would be a new project and we ask the reviewer to understand that this can no longer be part of the current manuscript.

Round 2

Reviewer 1 Report

Comments and Suggestions for Authors

Thank you for your responses